# How to Prune Your Language Model: Recovering Accuracy on the "Sparsity May Cry" Benchmark

Eldar Kurtic[1]*, Torsten Hoefler[2], Dan Alistarh[1,3]

[1]Institute of Science and Technology Austria, [2]ETH Zürich, [3]Neural Magic, Inc.

Pruning large language models (LLMs) from the BERT family has emerged as a standard compression benchmark, and several pruning methods have been proposed for this task. The recent "Sparsity May Cry" (SMC) benchmark put into question the validity of all existing methods, exhibiting a more complex setup where many known pruning methods appear to fail. We revisit the question of accurate BERT-pruning during fine-tuning on downstream datasets, and propose a set of general guidelines for successful pruning, even on the challenging SMC benchmark. First, we perform a cost-vs-benefits analysis of pruning model components, such as the embeddings and the classification head; second, we provide a simple-yet-general way of scaling training, sparsification and learning rate schedules relative to the desired target sparsity; finally, we investigate the importance of proper parametrization for Knowledge Distillation in the context of LLMs. Our simple insights lead to state-of-the-art results, both on classic BERT-pruning benchmarks, as well as on the SMC benchmark, showing that even classic gradual magnitude pruning (GMP) can yield competitive results, with the right approach.

## 1. Introduction

The massive growth of accurate language models (LMs) has motivated several advanced *model sparsification* techniques [1], encompassing unstructured and structured pruning. In this paper, we focus on the popular task of unstructured pruning of LMs [2–5], that is, removing individual weights from such models, which is known to lead to both storage and computational benefits [5].

The recent "Sparsity May Cry (SMC-Bench)" benchmark [6] investigates unstructured pruning of BERT-family models, specifically RoBERTa-large [7], on a set of more complex sequence-based tasks, e.g. CommonsenseQA [8] and WinoGrande [9].

The authors reach a very surprising conclusion about sparse neural networks (SNNs), namely:

> *"All of the SOTA sparse algorithms bluntly fail to perform on SMC-Bench, sometimes at significantly trivial sparsity e.g., 5%. [...] This observation alarmingly demands the attention of the sparsity community to reconsider the highly proclaimed benefits of SNNs."* [6]

**Contribution.** In this paper, we follow this call to arms. On the *constructive* side, we codify a consistent set of pruning best-practices for LLMs, which we either state for the first time, or we isolate as having been *implicitly or explicitly adopted* by the literature, e.g. [1]. On the *deconstructive* side, we examine the relationship between these best-practices and SMC-Bench, and show that adapting the benchmark's setup to follow these best-practices inverts the very strong negative claims made by SMC, even in the case of basic gradual magnitude pruning (GMP) [10]. Figure 1 provides an illustration of accuracy recovery across various sparsity targets after applying our guidelines on the hardest task in the SMC benchmark, as identified by its authors, with two different pruners, GMP and oBERT, with and without Knowledge Distillation (KD).

---

*Correspondence to: Eldar Kurtic <eldar.kurtic@ist.ac.at>, Dan Alistarh <dan.alistarh@ist.ac.at>.

First Conference on Parsimony and Learning (CPAL 2024).

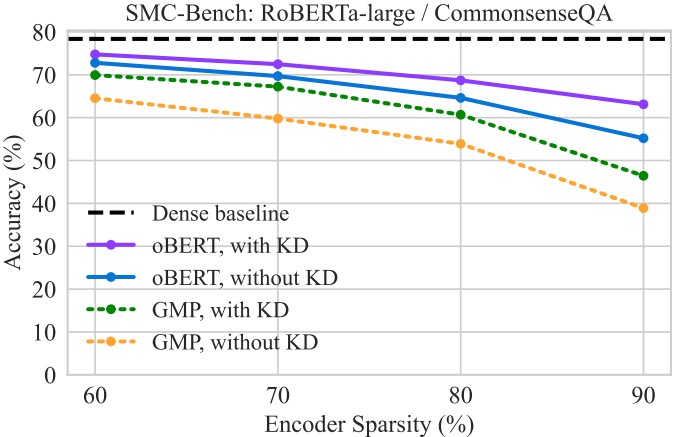

Figure 1: Accuracy recovery on the "hardest" task in the SMC-Bench suite after following the pruning guidelines we propose, using either basic gradual magnitude pruning (GMP) or the state-of-the-art second-order Optimal BERT Surgeon pruning (oBERT) [5].

The LLM pruning best-practices we propose are as follows:

1. *The length of the post-pruning training period, as well as the sparsification and learning rate schedules should be adapted to the desired target sparsity and model/task combination.*

2. *Certain model components, such as embeddings and classification heads, naturally have outsize impact on accuracy. Since pruning them brings negligible performance gains for Transformer models such as the ones considered in SMC-Bench, these layers should remain dense.*

3. *Knowledge distillation [11], which is known to bring significant gains even for dense models, should be standard for LM pruning, as it can be highly-effective when properly-tuned.*

**Contributions.** In this setting, our main contributions are as follows:

1. We detail and justify the above guidelines, both conceptually and practically, in the context of standard BERT-pruning benchmarks, widely adopted in the literature, on tasks such as question-answering on SQuADv1.1 [12], sentence classification Quora Duplicate Query Dataset QQP [13], and natural language inference MNLI [14].

2. We then instantiate them directly to the two tasks identified by the authors to be "hardest" by SMC-Bench, CommonsenseQA [8] and WinoGrande [9]. Across both settings, following these best-practices outperforms the best existing results on the benchmark, by wide margins at larger sparsities. Specifically, we are able to achieve high sparsities, in the range of 80-90%, accurately (see e.g. Figure 1) on SMC-Bench, contradicting the strongly negative claims initially made by this benchmark.

3. In conjunction with the accurate oBERT pruner, our setup sets new state-of-the-art sparsity-vs-accuracy results for the BERT-base model on the SQuADv1.1 task. Moreover, on SMC-Bench, both GMP and oBERT can provide stable results when pruning RoBERTa-large up to 90% sparsity. Our work therefore shows that unstructured sparsity can in fact be a viable option even in challenging settings.

## 2. Context

We examine a standard setting for LM pruning, in which models are first *pre-trained* on a large *upstream* corpus of unlabelled text. Then, they are *fine-tuned* in a supervised manner on a smaller *downstream* task, such as question-answering or text-classification. Specifically in this work, we focus

on *downstream pruning*, where pruning and fine-tuning are done directly on the target downstream dataset.

As a running baseline, we use *gradual magnitude pruning (GMP)* [10, 15], which periodically removes a fraction of weights with smallest magnitudes during training, interspersed with fine-tuning steps. However, the literature on pruning LLMs, and in particular BERT models [2–4], clearly states that GMP *does not* perform well, and uses this as motivation for more complex methods. We contradict this claim here, showing that well-tuned GMP can outperform results of most prior methods. As an alternative to GMP, we will also employ the currently state-of-the-art oBERT pruner [5]. For a fair comparison, we follow most prior references which focus on pruning BERT-base [16] on standard tasks such as SQuADv1.1 and a subset of the GLUE benchmark [17].

**The SMC Benchmark.** The recently-proposed "Sparsity May Cry" benchmark, which we alternatively call SMC or SMC-Bench, contains four categories of tasks: commonsense reasoning, arithmetic reasoning, protein thermostability prediction, and multilingual translation.

Of these, the first category, commonsense reasoning, is identified by the authors to be the most challenging; specifically, the CommonsenseQA (CSQA) [8] is stated to be the "hardest" task in terms of accuracy-vs-sparsity trade-offs, as all pruning methods appear to crash to random accuracy even at low (10-20%) sparsity. Due to space constraints, we will mainly focus on this CSQA task, but also validate our results on other tasks, such as WinoGrande.

On CSQA, SMC provides results for sparsities between 20% and 90% using a fixed-length schedule of 3-epochs with linearly-decaying learning rate, *which is the same as fine-tuning recipe for the dense pre-trained model*. The standard version of SMC prunes *all layers* except LayerNorm, including the *token, segment, and position embeddings, as well as the classification head*. Knowledge Distillation [11] is *not used* in SMC experiments.

## 3. Pruning Best Practices: A Case Study on BERT Architectures

### 3.1. What to prune?

**Matching Sparsity to Model Structure.** The multi-layer bidirectional BERT architecture [16] is comprised of three key components: an embedding component, an encoder, and a task-specific classification head. The embedding component, in turn, is composed of three sub-components: token embeddings, segment embeddings, and positional embeddings. These sub-components transform tokenized words into H-dimensional vector representations, where H refers to the model's hidden-dimension. Specifically, we have the following:

- Token embeddings map each token to its corresponding vector representation, which is then combined with the position and segment embeddings to form a unique representation for each input token.

- As BERT is a bidirectional model, positional embeddings encode positional information for each token, ensuring that tokens in different positions within a sentence are not represented with the same embedding vector.

- Segment embeddings encode information about multiple sequences packed together for downstream tasks (e.g. context-question in SQuAD, duplicate questions in QQP, etc).

These embeddings are learned through unsupervised methods, often using large text corpora, and they capture the distributional properties of words, enabling the model to infer meaning and relationships between them. Position embeddings address the sequential nature of language by providing positional information to the model. They encode the relative or absolute positions of words within a sentence, allowing the model to understand the contextual dependencies and capture important syntactic and structural information.

Table 1: A brief overview of FLOPs and parameter counts of the three main components of the RoBERTa-large model on the CSQA task from the SMC-Bench suite. For simplicity, we count FLOPs and parameters only for weights of all linear layers in the model and ignore negligible costs associated with LayerNorm, Dropout, biases, and non-linearities such as ReLU/GELU.

|  | Parameter count | Fraction of total | FLOPs | Fraction of total |
|---|---|---|---|---|
| Embeddings | 51.5M | 14.5% | $\approx 0$ | $\approx \mathbf{0\%}$ |
| Encoder | 302M | 85.5% | 604M | 99.9% |
| Classification head | 0.001M | $\approx \mathbf{0\%}$ | 0.002M | $\approx \mathbf{0\%}$ |

Conceptually, unstructured pruning of pre-trained embedding layers during short fine-tuning stage would completely destroy learned token representations, and literally make the model position-agnostic if a large fraction of positional encodings are sparsified.

**Practical Issues.** Virtually all popular frameworks, e.g. Hugging Face Transformers [18] and Py-Torch [19], implement embeddings as lookup tables, which clearly implies that imposing unstructured sparsity will not improve their efficiency. In addition, pruning of the classification head for models at the scale considered in SMC-Benchmark is not desirable, since at higher sparsities entire rows/columns could get pruned, thus disabling model's ability to ever predict those logits. To formalize these arguments, in Table 1 we present FLOP and parameter count analysis which demonstrate that these two components, embeddings and classifier head, consume negligible amount of compute. While pruning embeddings could be justified in terms of reducing parameter count (but not computational cost!), pruning the classification head is hard to justify via either criterion.

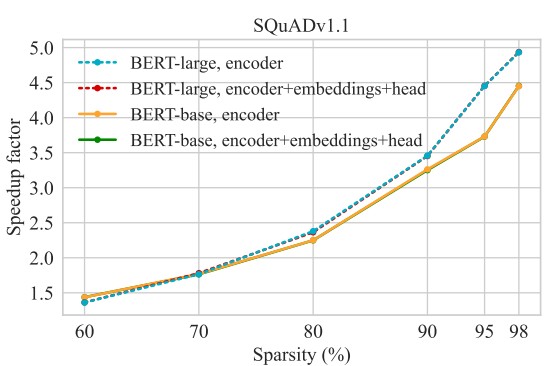

Figure 2: Speedups of sparse BERT-base and BERT-large models relative to corresponding dense baselines, evaluated in the sparsity-aware CPU-inference engine DeepSparse (version 1.4) at 4-cores of AMD EPYC 7702, batch-size 32, and sequence-length 384. Inference speedups from sparse embeddings and classification head (encoder+embeddings+head) are negligible, hence the plots are perfectly superimposed with plots where only the encoder is sparsified.

Figure 3: Sensitivity analysis of the impact of pruning the embeddings and the classification head on accuracy of the fine-tuned BERT-base model on the SQuADv1.1 dataset. Both GMP and oBERT pruners, suffer from significant performance drops even at moderate sparsities. Accuracy of GMP pruned model deteriorates quickly, even at sparsities as low as 40%, whereas oBERT absorbs negative impacts from pruning of embeddings and head until 60% sparsity target.

**Cost-Benefit Analysis.** To examine the effects of pruning the embeddings and classification head we perform a simple cost-vs-benefits analysis. First, we sparsify either encoder alone or altogether encoder, embeddings and classification head in BERT-base and BERT-large models. On the one hand, we evaluate the speedups of the resulting sparse models in the sparsity-aware CPU-inference engine DeepSparse [20]. Figure 2 demonstrates that there are no speedup improvements when embeddings and head are sparsified, *even at 98% sparsity*. Second, in Figure 3 we examine the drops in accuracy when these layers are included in pruning to reach target sparsities, a procedure also known as sensitivity analysis [21]. GMP starts dropping accuracy even at 40% sparsity target, while

oBERT manages to absorb impacts until 60%, after which the drops are critical and the models become unusable in practice. In summary, this analysis shows that, in this context, pruning the embeddings and classification head does not yield practical gains during inference, while negatively impacting model's accuracy. Therefore, we suggest to keep these layers dense and prune only the encoder, where the majority of the computational gains can be made.

We note that not pruning embeddings is a well-established practice in the LLM-pruning literature [1–5, 22–24], as illustrated in Table 6. **We identify this choice as the first major cause explaining the apparent failure of pruning methods on the SMC benchmark.**

### 3.2. The Impact of Knowledge Distillation

Knowledge Distillation (KD) is standard in many pruning references [2, 4, 5, 22, 23]. The loss function is formulated as a linear combination of the standard loss associated with the specific task (e.g. cross-entropy for classification $\mathcal{L}_{CE}$) and the KL divergence ($\mathcal{L}_{KL}$) between output distributions of the dense (teacher) model and the sparse (student) model in the form: $\mathcal{L} = (1-h)\mathcal{L}_{CE} + h\mathcal{L}_{KL}$. The ratio between the two is controlled with the *hardness* hyperparameter $h$. To determine its optimal value at high sparsities we run an ablation study (Table 4), and adopt value $h = 1$.

**Knowledge Distillation Temperature.** The temperature $T$ is an additional KD-hyperparameter that requires proper tuning, as it controls the "softness" of the output distribution. In the pruning literature, it is standard to use the "stronger" $T = 1$ or $T = 2$ values [2, 4, 5, 22, 23]; we revisit this by visualizing teacher's output distributions to get an insight into what the sparse student is learning.

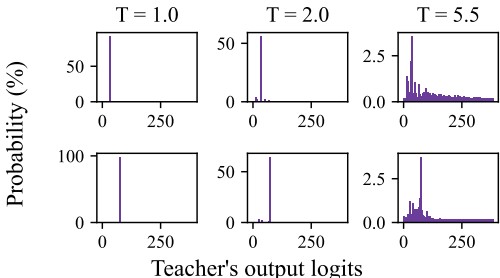

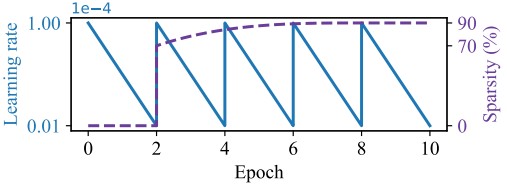

Figure 4: Teacher's output distribution at commonly used temperatures $T \in \{1.0, 2.0\}$ and the proposed $T = 5.5$.

Figure 5: Visualization of the learning rate with rewinds and accelerated cubic sparsity scheduler for the proposed gradual pruning framework.

In Figure 4, we visualize generated distributions for randomly picked samples from the SQuADv1.1 task softened with three values of the temperature. As can be seen, teacher's high confidence in predicting the correct class at the commonly used temperatures $T \in \{1.0, 2.0\}$ makes the knowledge distillation almost obsolete. Motivated by this observation, we run an ablation study for many higher temperatures and report a fraction of results in Table 5. Given the results, we adopt the temperature $T = 5.5$.

### 3.3. How to Prune? The Importance of the Pruning Schedule

As noted in the Background section, SMC adopts a fixed gradual pruning schedule, independently of the target sparsity, and observes that all known methods quickly collapse even on moderate sparsities, around $50\% - 60\%$, on CSQA task.

We probe the validity of this choice of scheduling via the following experiment. We apply the state-of-the-art oBERT pruner *in One-Shot*, without any retraining, on the RoBERTa-large / CSQA SMC-Bench task, while allowing the pruner to sparsify embeddings and classification head. This makes the setup much harder for pruning but enables a fair comparison against results presented in the SMC-Bench.

In Figure 6 we see that even in this setup, *without any fine-tuning*, this one-shot pruning approach *does not collapse* up to 80% sparsity, while also *significantly outperforming all other methods*, which have been executed in *gradual pruning* fashion, with short fine-tuning cycles to recover from accuracy drops incurred by pruning. These results, in which One-Shot pruning significantly outperforms gradual pruning across different methods, point to the need for *careful design of gradual pruning schedules*, such that the fine-tuning cycles actually enable accuracy recovery. **We identify this as a second cause for failure of pruning methods on the SMC benchmark.**

We now take a step back, and reflect upon the most important components of a gradual pruning schedule, which needs to be designed to recover from pruning accuracy drops.

**Accelerated sparsity schedule.** The established convention adopted in the literature is to impose sparsity on the model by following either a cubic [15] or a linear sparsity scheduler, starting from the dense model (zero-sparsity) and pruning until the target sparsity is reached. Motivated by the observation that language models are heavily overparametrized for downstream tasks, we emphasize the importance of a seemingly-incremental improvement in the aforementioned sparsity schedulers. Namely, a large first pruning step turns out to be of a crucial importance for competitive results at high target sparsities (e.g. 97%). Instead of starting to prune the model from zero-sparsity, in the first pruning step the model should be pruned to a much higher target (e.g. 50% or 70%). This leaves more time to distribute pruning to high sparsity targets over a longer fine-tuning range, and thus enables better accuracy recovery. In Table 2, we report results from an ablation study with respect to the size of the initial pruning step on the standard benchmark. Removing either 50% or 70% of weights in the initial step significantly helps improving results at very high sparsity (97%). We visualize the accelerated sparsity scheduler in Figure 5.

Table 2: Ablation study for initial pruning step on the BERT-base/SQuADv1.1 benchmark.

| Sparsity (%) | F1 score at sparsity target | |
|---|---|---|
| | 90% | 97% |
| 0 | 85.2 | 77.2 |
| 30 | 85.5 | 77.8 |
| 50 | **85.8** | 78.5 |
| 70 | 85.8 | **79.1** |

Table 3: Ablation study for initial learning rate on the BERT-base/MNLI benchmark.

| Initial LR | Accuracy at sparsity target | |
|---|---|---|
| | 90% | 97% |
| 3e-5 | 80.8 | 76.3 |
| 5e-5 | 81.4 | 77.8 |
| 8e-5 | **81.9** | 78.6 |
| 1e-4 | 81.6 | **79.3** |

Table 4: Ablation study for Knowledge Distillation hardness on the BERT-base/SQuADv1.1 benchmark.

| Hardness | F1 score at sparsity target | |
|---|---|---|
| | 90% | 97% |
| 0.6 | 84.6 | 78.4 |
| 0.8 | 85.9 | 80.1 |
| 0.9 | 86.2 | 80.7 |
| 1.0 | **86.7** | **81.0** |

Table 5: Ablation study for Knowledge Distillation temperature on the BERT-base/SQuADv1.1 benchmark.

| Temperature | F1 score at sparsity target | |
|---|---|---|
| | 90% | 97% |
| 1.0 | 84.7 | 77.3 |
| 2.0 | 85.8 | 79.0 |
| 5.5 | **86.7** | **81.0** |
| 8.5 | 86.4 | 80.9 |

**Learning rate schedule.** Our goal is to provide a simple baseline setup that works well across wide range of datasets without any additional task-dependent tuning. Currently, papers either report best results following an extensive hyperparameter search for each task, e.g. Zafrir et al. [4] investigate more than 50 different hyper-parameter settings, or they make use of carefully crafted schedulers for each setup independently which may include warm-up phases with and without rewinds [2, 5]. This may lead to high specialization to the target task/model, which is undesirable in practice and makes it hard to distinguish benefits from the pruning technique itself. We propose to simply *replicate* the standard dense fine-tuning schedule [16] by a certain factor and intertwine it with pruning steps. For a fair comparison with Sanh et al. [2] we replicate the 2-epoch fine-tuning schedule by a factor of 5, matching their 10-epoch setup. For a fair comparison with Chen et al. [3] we replicate it

by a factor of 15, reproducing their 30-epoch setup. For convenience, we visualize the learning rate schedule in Figure 5. In Appendix B, we describe results with inferior schedulers.

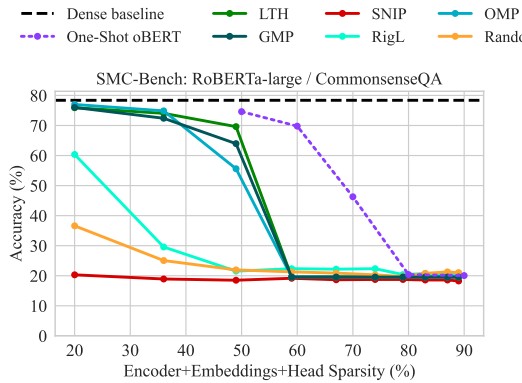

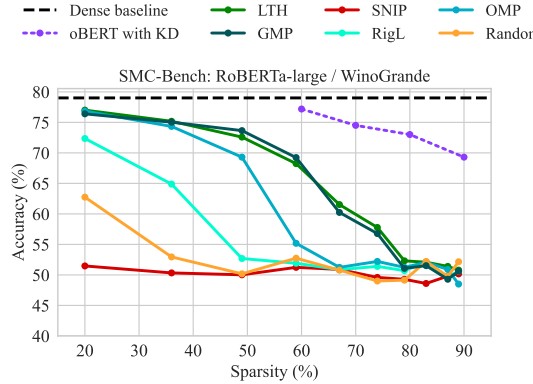

Figure 6: For comparison with SMC-Bench results, we prune encoder, embeddings and classifier head of the model in **One-Shot, without any retraining**, with second-order oBERT pruner. Relative to all other methods, which were applied by SMC-Bench in a gradual fashion with retraining phases in-between pruning steps, oBERT demonstrates dominance up to very high sparsities (<80%).

Figure 7: Accuracy recovery on the second "hardest" task in the SMC-Bench suite, after following our proposed pruning guidelines with the oBERT pruner. Results of other methods are obtained by the SMC-Bench work, and are not a direct comparison since oBERT prunes only the encoder while others distribute sparsity over encoder, embeddings, and classifier head, resulting in a more dense encoder which leads to reduce inference speed (see Figure 2).

### 3.4. Literature Context

To reinforce our points from the previous sections regarding what and how to prune, we summarize the set of choices made by some representative methods from the literature, in Table 6. It appears clear that the literature largely follows our best practices in terms of choice of *what to prune*, which appears reasonable given our discussion on practicality above.

Table 6: Overview choices for pruning made by Sparsity May Cry benchmark, relative to choices made by the most representative literature on LM pruning over time.

| | What to prune? | | | How to finetune? | |
|---|---|---|---|---|---|
| | Embeddings | Encoder | Classification head | Extended schedule | Knowledge distillation |
| Movement Pruning [2] | No | Yes | No | Yes | Yes |
| Lottery Tickets [3] | No | Yes | No | Yes | No |
| Block Movement Pruning [23] | No | Yes | No | Yes | Yes |
| Sparse BERT [22] | No | Yes | No | Yes | Yes |
| Prune Once for All [4] | No | Yes | No | Yes | Yes |
| Optimal BERT Surgeon [5] | No | Yes | No | Yes | Yes |
| PLATON [24] | No | Yes | No | Yes | No |
| Sparsity May Cry [6] | Yes | Yes | Yes | No | No |

## 4. Experimental Validation

Now, we aggregate all of the previously analyzed improvements in a *downstream pruning recipe*, which we summarize for convenience in Appendix A. We validate its effectiveness on two benchmarks: the standard BERT-base pruning benchmark widely adopted across the pruning literature [2–5, 23], and the recently proposed Sparsity May Cry (SMC) benchmark [6].

## 4.1. Results on the standard BERT-base benchmark

The standard BERT-base benchmark consists of pruning the BERT-base model on three downstream datasets: extractive question-answering SQuADv1.1 [12], sentence classification Quora Duplicate Query Dataset QQP [13], and natural language inference MNLI [14]. Pruning refers to the process of removing weights (connections) from all linear layers in the encoder part of the BERT architecture, which amounts to 85M params out of the total 110M. All sparsities are reported with respect to the encoder size. Pruning is performed in a gradual manner where pruning steps are intertwined with fine-tuning steps to recover accuracy.

**Improved gradual magnitude pruning (GMP⋆).** To illustrate effectiveness of the proposed best practices, we first focus on improving the classic gradual magnitude pruning approach, which we simply call GMP⋆. The literature on BERT-pruning [2–4] motivates development of new pruning techniques by demonstrating very poor performance of the baseline GMP technique. In this section we argue that such negative conclusions arise mainly due to the poorly designed pruning and fine-tuning schedules, and can be mitigated by adopting our best practices. In Table 7 we present downstream pruning results obtained with our GMP⋆ and other GMP-based baselines. For a fair comparison with respect to the compute budget, we consider both setups, 10- and 30-epoch. In the former, we compare against the GMP baselines reported in Sanh et al. [2] and refer to them as GMP_{MvP}. In the latter, we compare against the best results in Chen et al. [3], obtained either via GMP or Lottery Ticket (LTH) approach, and refer to them as GMP_{LTH}. As can be seen from the Table 7, our GMP⋆ remarkably outperforms all other results by *extremely large margins*; in some cases by even more than **20 points**! These results indicate a significant improvement in the performance of the GMP baseline, elevating it to a level of competitiveness that rivals top-performing pruners.

Table 7: Downstream pruning comparison of GMP⋆ with other GMP-based baselines. GMP⋆ remarkably outperforms all other approaches by extremely large margins.

| Method | Spars. | Ep. | SQuAD F1 | MNLI m-acc | QQP acc |
|---|---|---|---|---|---|
| BERT-base | 0% | | 88.5 | 84.5 | 91.1 |
| GMP_{MvP} | 90% | 10 | 80.1 | 78.3 | 79.8 |
| GMP⋆ | | | **86.7** | **81.9** | **90.6** |
| GMP_{MvP} | 97% | 10 | 59.6 | 69.4 | 72.4 |
| GMP⋆ | | | **81.3** | **79.1** | **89.7** |
| GMP_{LTH} | 90% | 30 | 68.0 | 75.0 | 90.0 |
| GMP⋆ | | | **87.9** | **82.7** | **90.8** |
| GMP⋆ | 97% | 30 | 85.4 | 80.9 | 90.6 |

Table 8: Downstream pruning comparison of GMP⋆ with advanced pruning techniques.

| Method | Spars. | Ep. | SQuAD F1 | MNLI m-acc | QQP acc |
|---|---|---|---|---|---|
| BERT-base | 0% | | 88.5 | 84.5 | 91.1 |
| GMP⋆ | 90% | 10 | **86.7** | **81.9** | **90.6** |
| MvP | | | 84.9 | 81.2 | 90.2 |
| GMP⋆ | 97% | 10 | 81.3 | 79.1 | **89.7** |
| MvP | | | **82.3** | **79.5** | 89.1 |
| GMP⋆ | 90% | 30 | 87.9 | 82.7 | 90.8 |
| oBERT | | | 88.3 | **83.8** | **91.4** |
| oBERT⋆ | | | **88.6** | | |
| GMP⋆ | 97% | 30 | 85.4 | 80.9 | 90.6 |
| oBERT | | | 86.0 | **81.8** | **90.9** |
| oBERT⋆ | | | **86.6** | | |

We now compare GMP⋆ with methods that rely on higher-order information to make pruning decisions, like gradients in Movement Pruning (MvP) [2] and the loss curvature in oBERT [5]. Both of these have higher computational overhead, but we still put our results in context to realize the extent of the improvements introduced by following the above guidelines. As can be seen from results in Table 8, GMP⋆ is able to improve upon the performance of MvP in 4 out of 6 configurations, but cannot match the performance of the oBERT method. In addition to these comparisons, we run the state-of-the-art BERT-pruning method oBERT with optimized hyperparameters from GMP⋆ on the SQuADv1.1 task. We refer to these results as oBERT⋆. As can be seen from the Table 8, even the very competitive oBERT results benefit from the GMP⋆ setup. For all GMP⋆ runs, we report mean performance across three runs with different seeds.

Since oBERT presented state-of-the-art results on BERT-base/SQuADv1.1 benchmark, we observe that following our guidelines (oBERT⋆) leads to new state-of-the-art pruning results on this benchmark.

### 4.2. Results on the SMC Benchmark

Now, we adapt our downstream pruning recipe to the RoBERTa-large model and conduct experiments on the two "hardest" tasks in the SMC Benchmark, unstructured pruning of the RoBERTa-large model on CommonsenseQA and WinoGrande datasets. Specifically, we adopt the previously presented best practices as follows: do not prune the embedding layer and classification head (Section 3.1), scale the fine-tuning schedule for better accuracy recovery according to Section 3.3, and use well-tuned Knowledge Distillation (Section 3.2).

Figure 1 shows that, contrary to what has been reported in Liu et al. [6], basic gradual magnitude pruning (GMP) does not fail to perform on SMC-Bench. In addition to GMP results, the figure demonstrates that approximate second-order information via oBERT ensures even better accuracy recovery in this challenging setup. Further, we demonstrate that incorporating a properly-configured Knowledge Distillation on top of the gradual pruning schedules brings non-trivial improvements in accuracies for sparse models.

To demonstrate that our proposed guidelines do not pertain only to the CommonsenseQA dataset presented in Figure 1, we report results on the second "hardest" task (WinoGrande) of the SMC Benchmark in Figure 7. Namely, we apply second-order oBERT pruning in conjunction with our proposed gradual pruning guidelines. As can be seen from the Figure, even in this task, gradual pruning does not bluntly fail to perform and reasonably recovers accuracy of the dense model relative to all other methods reported in Liu et al. [6] which at 80% produce unusable models.

## 5. Conclusion

We have presented, discussed and evaluated a set of three simple guidelines for successful pruning of LLMs in the "downstream" pruning setup, illustrating our results on BERT-base and RoBERTa-large models, across different tasks, including the recent SMC benchmark. Some of the best-practices we proposed are either known or implicitly-adopted by some, but not all, works in the literature.

As we have discussed and shown experimentally, following these simple guidelines can lead to much more solid baseline performance across different models and tasks. Specifically, the fact that our variant of GMP⋆ outperforms most of the existing methods in the literature, and is the first method to provide good performance on SMC-Bench suggests that there is value to popularizing these best-practices. Further, our work provides a solid set of *new competitive baselines*, which can help researchers interested in high-performance compression of large language models.

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

# A. Downstream pruning recipe

All of our implementations are built on top of HuggingFace's Transformers [2] [18] and Datasets [3] [25] libraries, and NeuralMagic's SparseML [4] [26] library for model compression, and will be open-sourced to community along with our sparse models.

As our goal is to provide a simple and unique gradual pruning setup, all of our downstream runs (for all datasets) are using the same set of hyperparameters. The ones used to obtain results reported in Tables 7 and 8, are as follows:

- `learning-rate`: recurring 2-epoch scheduler (visualized in Figure 5) with the initial value of `1e-4`, and the final value of `1e-6`,
- `number-of-epochs`: 10 or 30 epochs, depending on the methods we compare against,
- `sparsity`: cubic scheduler with the initial pruning step of 70% sparsity (visualized in Figure 5),
- `pruning`: prune frequency of ten times per epoch, except during the first and last 2-epochs when only fine-tuning happens and masks are fixed,
- `student-initialization`: standard BERT-base(`bert-base-uncased`[5]),
- `knowledge-distillation (KD)`: (hardness, temperature) = (1.0, 5.5),
- `KD-teachers`: standard BERT-basefine-tuned on the corresponding task,
- `weight-decay`: 0.0,
- all other hyper-parameters are set to the standard default values, e.g. Sanh et al. [2]:
    - SQuADv1.1: `batch-size=16`, `max-sequence-length=384`, `doc-stride=128`,
    - MNLI and QQP: `batch-size=32`, `max-sequence-length=128`.

# B. Learning rate schedulers we tried, but didn't work

The schedulers we tried but didn't work: 1) linearly decaying learning rate, 2) the default fine-tuning learning rates (3e-5 for SQuADv1.1 and 2e-5 for MNLI and QQP), 3) learning rates with the warm-up phase. In the preliminary experiments, we have noticed that 1) and 2) have problems in recovering from the pruning steps at higher sparsities. The former one has extremely small learning rate values during the last few epochs when the model is pruned to high sparsities. The latter one continuously fails to recover properly even at moderate sparsity targets, which is why we run a sweep over a range of initial learning rate values. Given the results in Table 3, we decided to proceed with the 1e-4 as it helped to recover significantly at high sparsities. We haven't observed any benefits from the warmup phase, which is why we have decided not to use it as it adds an additional hyperparameter to tune.

---

[2]https://github.com/huggingface/transformers
[3]https://github.com/huggingface/datasets
[4]https://github.com/neuralmagic/sparseml
[5]https://huggingface.co/bert-base-uncased

