# OpenReview forum: "How to Prune Your Language Model: Recovering Accuracy on the ``Sparsity May Cry'' Benchmark"
_CPAL.cc/2024/Conference — CPAL 2024 (Proceedings Track) Oral_

### Meta-Review · Area_Chair_JstC · 2023-11-10

**Recommendation:** Accept (Oral)
**Confidence:** 5

**Metareview:**

(this is an invited paper, so only one meta-review is provided)

The paper addresses the crucial topic of unstructured pruning in language models (LMs) with a particular focus on the recently proposed "Sparsity May Cry (SMC-Bench)" benchmark. The authors respond to SMC-Bench's surprising conclusion that state-of-the-art (SOTA) sparse algorithms fail consistently, even at low sparsity levels. SMC poses a new challenging benchmark for the community, especially for large-scale settings. This paper comprehensively studies the BERT pruning and provides a set of general guidelines for successful pruning on SMC-Bench. The main message delivered by this paper is that following a set of three guidelines, simple gradual magnitude pruning (GMP) can yield competitive results. Overall, the authors have done an excellent job of summarizing the principles for BERT-pruning. The contribution of this paper is timely and significant for the community.

Contributions and Strengths:
1.	Pruning Best Practices: The authors make a significant contribution by proposing a set of pruning best practices for BERT-pruning. These guidelines, presented for the first time or refined from existing literature, include considerations for the post-pruning training period, sparsification components, learning rate schedules, and knowledge distillation.
2.	Deconstruction of SMC-Bench: The paper meticulously analyzes the relationship between the proposed best practices and the SMC-Bench benchmark. By applying their guidelines to the benchmark's setup, the authors challenge the negative claims made by SMC-Bench, even for basic gradual magnitude pruning (GMP). The authors support their claims with concrete and convincing results.
3.	In-depth Analysis: I like the analysis of the efficiency analysis of different components in RoBERTa, which provides convincing and clear guidance for practitioners and researchers. The extensive ablation study in Table 2-5 is also rich and useful.
4.	Performance on Hardest Tasks: The authors demonstrate the effectiveness of their proposed guidelines on the tasks identified by SMC-Bench as the "hardest" – CommonsenseQA and WinoGrande. Across both settings, their approach outperforms existing results on the benchmark, achieving high sparsities of 80-90% accurately.
5.	State-of-the-Art Results: The paper claims new state-of-the-art sparsity-vs-accuracy results for the BERT-base model on the SQuADv1.1 task. Moreover, their approach, coupled with the oBERT pruner, achieves stable results when pruning RoBERTa-large up to 90% sparsity, challenging the notion that unstructured sparsity is not viable in challenging settings.

Overall, the paper makes a valuable contribution to the field of BERT-pruning by proposing effective best practices and challenging the findings of the SMC-Bench benchmark. This paper, together with SMC-Bench, provides a clearer picture of whether pruning a larger language model is easier or harder, summarizing the overall progress of pruning in the context of BERT. It would be a valuable and welcome contribution to CPAL.

---

### Decision · Program_Chairs · 2023-11-20

Accept (Oral)